# KBeagle: An Adaptive Strategy and Tool for Improving Imputation Accuracy and Computation Time

**DOI:** 10.3390/ijms26125797

**Published:** 2025-06-18

**Authors:** Xingyu Guo, Jie Qin, Shikai Wang, Jincheng Zhong, Li Liu, Yixi Kangzhu, Daoliang Lan, Jiabo Wang

**Affiliations:** 1Key Laboratory of Qinghai-Tibetan Plateau Animal Genetic Resource Reservation and Utilization, Ministry of Education and Sichuan Province, Southwest Minzu University, Chengdu 610041, China; 230710002010@stu.swun.edu.cn (X.G.); qj598166864@163.com (J.Q.); 220710002015@stu.swun.edu.cn (S.W.); 22100047@swun.edu.cn (J.Z.); 23900003@swun.edu.cn (Y.K.); 2Key Laboratory of Combining Farming and Animal Husbandry of Ministry of Agriculture, Institute of Animal Husbandry, Heilongjiang Academy of Agricultural Sciences, Harbin 150028, China; llxmrn@haas.cn

**Keywords:** imputation, Beagle, K-means, cluster, genome sequencing

## Abstract

Whole-genome sequencing (WGS) technology has made significant progress in obtaining the genomic information of organisms and is now the primary way to uncover genetic variation. However, due to the complexity of the genome and technical limitations, large genome segments remain ungenotyped. Imputation is a useful strategy for predicting missing genotypes. The accuracy and computing speed of imputation software are important criteria that should inform future developments in genomic research. In this study, the K-Means algorithm and multithreading were used to cluster reference individuals to reduce the number and improve the length of haplotypes in the subpopulation. We named this strategy “KBeagle”. In the comparison test, we determined that the KBeagle-imputed dataset (KID) can identify more single-nucleotide polymorphism (SNP) loci associated with the specified traits compared to the Beagle-imputed dataset (BID), while also achieving much lower false discovery rates (FDRs) and Type I error rates under the same power of detection of association signals. We envision that the main application of KBeagle will focus on livestock sequencing studies under a strong genetic structure. In summary, we have generated an accurate and efficient imputation method, improving the imputation matching rate and calculation time.

## 1. Introduction

With the rapid advancement of high-throughput genotyping technologies, whole-genome sequencing has emerged as the predominant method for identifying heritable variations and elucidating genomic architecture. Nevertheless, missing genotypes remain prevalent in whole-genome analyses, which is attributable to various factors such as technical limitations in sequencing, complex genomic structures, and variable sample quality. Because there is a tight linkage disequilibrium (LD) phenomenon, the markers in common LD blocks share a similar genotype. Through the linkage relationship between markers, the missing genotype can be imputed. Genotype imputation serves as a robust computational approach that mitigates the limitations of low-density marker datasets by inferring missing genotypes. This technique operates on the principle of utilizing known genotype data from reference panels to predict probable genotypes in target samples, primarily by exploiting the linkage information surrounding the missing loci [1].

Current genotype imputation tools can be broadly classified into two categories based on computational efficiency: computation-intensive and computation-efficient approaches. Computation-intensive methods utilize all available genotype data to infer missing genotypes, achieving higher accuracy at the cost of increased computational time. Prominent examples of such software include IMPUTE5 [2], MaCH [3], and fastPHASE [4]. In contrast, computation-efficient imputation tools prioritize speed by restricting analysis to genotypes adjacent to the target single-nucleotide polymorphisms (SNPs), thereby reducing runtime while compromising slightly on accuracy. Widely used software in this category include PLINK [5] and Beagle [6]. Most tools, such as Minimac4, based on the Markov Chain Haplotyping (MaCH) algorithm, and IMPUTE5, which used the Positional Burrows–Wheeler Transform (PBWT) algorithm, rely on the physical positions or order of markers to model LD, requiring reference genome information. For species lacking a reference genome or those with highly complex genomic structures including frequent translocations/inversions, the effectiveness of these methods is limited.

Beagle stands as the most widely used imputation software, specializing in haplotype phasing and missing genotypes imputation. Its popularity stems from two key advantages: its lower memory consumption and its faster computational speed compared to alternative imputation tools [7]. Originally developed for human genetics, Beagle has since been extensively applied in genotype imputation studies across diverse species, including cattle [8,9], pigs [10], chickens [11], and other animals. Compared to human populations, animal breeding populations often exhibit more complex population stratification such as highly differentiated subpopulations and denser pedigree structures characterized by deeper lineages and larger family sizes. When dealing with highly stratified populations, Beagle’s Hidden Markov Model (HMM)-based algorithm may fail to effectively correct biases introduced by population stratification. For populations with exceptionally complex genetic structures containing those under recent strong selection or with high levels of inbreeding, its haplotype-clustering initialization strategy may prove inadequate. Additionally, as the size of the reference panel increases to include more than tens of thousands of samples, Beagle’s computational efficiency declines significantly, with memory usage rising sharply, limiting its applicability to ultra-large-scale datasets. To address this issue, clustering-based approaches have been proposed to account for genetic background differences. These methods partition datasets into genetically homogeneous clusters, where samples within each group exhibit high similarity while remaining distinct from other clusters [12]. As reported by Stachowicz et al. [13], imputation accuracy improves when reference panels and target samples share strong genetic correlations. Clustering ensures that highly related samples are grouped together, facilitating longer shared haplotype fragments and thus enhancing imputation precision. A subgroup with similar genetic information shares more haplotype fragments, resulting in the subgroup having longer haplotype fragments than the whole population. A clear population structure is a key factor for improving imputation accuracy. But this strategy also increases the computing time by separating the whole population and individually calculating each subpopulation.

In this study, we developed an imputation software named KBeagle, with the K-means clustering algorithm and multithread operation strategy. By comparing the accuracy and computing time of the newest version of Beagle (Beagle v5.4) and the KBeagle method, we hope to present its advantages and characteristics. Genomic prediction (GP) and a genome-wide association study (GWAS) involving the simulation phenotype were performed to validate the availability of imputed data. Successful application of KBeagle will be sufficient for meta-analysis, discovering new functional genes, and breeding using resequencing.

## 2. Results

### 2.1. Imputation Efficiency

Two copies of the data were used to test KBeagle’s imputation efficiency: a dataset of 354 Ashidan yaks (homogeneous data) and a dataset of 80 yaks of three breeds (structured data). The test results of the homogeneous data are shown in Figure 1A–D. The overall imputation accuracy of KBeagle is slightly lower than that of Beagle, and the imputation accuracy of KBeagle is unstable. Additionally, the imputation accuracy of KBeagle and Beagle will decrease with the increase in the missing rate and with the increase in the dataset size. However, the results of the structured data were different from the results of the homogeneous data. The structured data showed (Figure 1E–H) that the imputation accuracy of KBeagle was higher than that of Beagle, showing better imputation accuracy when the dataset was larger (50 K and 70 K).

We compared the computation time of KBeagle and Beagle across different SNP subsets (10 K, 30 K, 50 K, and 70 K) and varying missing rates (0.05, 0.1, 0.15, and 0.2) (Figure 1I–L). In all comparisons, Beagle consistently required more time for imputation than KBeagle, with computation time increasing as the missing rate increased, which was particularly evident in the 10 K and 70 K SNP subsets. In contrast, KBeagle exhibited a more stable performance within different missing rates. In summary, KBeagle demonstrated a lower computation time and greater computational stability than Beagle under different SNP subsets and varying missing rates.

### 2.2. Application in a Genome-Wide Association Study

We generated simulated phenotypes using GAPIT by specifying key genetic parameters with setting heritability and quantitative trait nucleotides (QTNs) to produce both continuous phenotypic distributions. The QTNs were randomly selected across the genome with effect sizes sampled from a standard normal distribution. By performing GWAS 60 times with simulated traits, we calculated the power, type I error, and false discovery rates (FDR) using three datasets: the non-missing dataset (ND), the Beagle-imputed dataset (BID), and the KBeagle-imputed dataset (KID). In the homogeneous Ashidan yak population data, KBeagle showed comparable performance to Beagle, with KID outperforming BID in 60 replicates for FDR control and 60 replicates for type I error (Appendix A). However, in the more structured three-breed yak data, KID demonstrated significantly improved performance in 60 replicates. KID was able to reduce the number of signals that were incorrectly associated with the specified traits to a greater extent than BID in one-time GWAS (Figure 2A). Based on the statistics of 60 replicates, under the same power condition, KBeagle also provided less FDR and Type I error (Figure 2B).

### 2.3. Application in Genomic Prediction

To further validate downstream analysis after imputation, GP was conducted for ND, BID, and KID. We generated simulated phenotypes using 5-fold cross-validation and GAPIT with specific key genetic parameters. The results revealed minimal differences in heritability and prediction accuracy among the three data types within the homogeneous Ashidan yak population (Appendix A). When examining yak data across the three breeds, the prediction accuracy remained consistent across ND, BID, and KID (Figure 3B). However, a notable distinction emerged in terms of the genetic influence on traits: genes in BID exhibited a stronger association with traits compared to those in KID. Additionally, the heritability estimates from BID more closely approximated those derived from real genotypes, suggesting that BID may provide a more reliable representation of true genetic effects (Figure 3A).

## 3. Discussion

Imputation of missing genotypes is an essential step in genome research and analysis. Beagle, one of the most commonly used imputation tools in plant and animal breeding, has been frequently updated, with the latest version, Beagle 5.4 [14], released in 2021. Since its first publication, Beagle has been extensively studied and utilized by many researchers. In a study by Nothnagel et al. [15], BEAGLE v3.0.1 was found to be more user-friendly than IMPUTE v0.3.2 and generally required less memory. Several scholars have compared different versions of Beagle, showing that the imputation quality of BEAGLE 5.0 and 5.1 surpasses that of BEAGLE 4.0 and 4.1, delivering good performance on livestock and crop datasets with minimal parameter adjustments [16]. When family information is absent in a cattle dataset, and parental and grandparent genotypes are excluded, the CPU time of Beagle 3.3.2 is significantly superior to that of Impute2, even with varying reference panel sizes [17]. Furthermore, the imputation of real sequencing data using iBLUP in hybrid pig populations demonstrated robust tolerance for high deletion rates, especially for rare variants [18]. This study primarily aims to improve the imputation of large genotypic datasets, where Beagle shows notable advantages in handling big data.

In this study, we compared two types of genotype data: one dataset related to an Ashidan yak population with a similar genetic background, and the other one contained data on a mixed population with three major breeds. Our results indicate that KBeagle outperformed Beagle in terms of imputation accuracy for the distant relationship population, but performed similarly in the inbred population. This finding aligns with the work of Steven G. Larmer et al. [19], who noted that clustering of highly correlated samples enhances imputation accuracy. A study by Korkuć et al. [20] showed that inter-sample correlation positively affected imputation accuracy in a German black-spotted cow dataset.

While closely related samples from a large population may share genetic similarities, the imputation accuracy of distantly related samples is typically lower due to Beagle’s inability to cluster highly correlated samples. Several studies have demonstrated that applying clustering strategies prior to imputation can significantly improve accuracy, particularly in genetically diverse or structured populations. For example, Zhang et al. [21] and Zhang et al. [22] showed that partitioning data into homogeneous subgroups before imputation enhances the local similarity within clusters, thereby improving genotype prediction. Li et al. [23] and Rahman and Islam [24] further explored fuzzy clustering methods, which address the limitations of hard partitioning by allowing individuals to share membership across multiple clusters. These approaches have proven especially valuable in contexts where population boundaries are ambiguous. However, this strategy has yet to be applied to completed software. In this study, we incorporated a silhouette-guided K-means clustering step in the KBeagle method, enabling adaptive subgroup detection and enhancing imputation performance. This addition aligns with and extends prior work by showing that data-driven clustering enhances haplotype sharing and imputation accuracy in complex genetic backgrounds. In contrast, KBeagle clusters these samples more effectively, allowing subpopulations with similar genetic information to share longer haplotype fragments, which leads to better imputation accuracy.

Interestingly, KBeagle showed better imputation results in populations with complex genetic structures, but similar accuracy in inbred populations. This suggests that KBeagle is more effective for datasets with intricate genetic relationships. However, the accuracy of imputation relies on the degree of fusion between LD and identity by descent (IBD) information. We hypothesize that the structure of genotype data influences clustering results, which in turn impacts imputation accuracy. Given that missing positions are artificially selected and the missing rates of some markers are higher, this may also affect the overall results. While the K-Means algorithm identifies the most appropriate cluster groups, these clusters do not always yield the highest mean imputation matching rates. Optimizing KBeagle would require finding the optimal number of clusters (K) that maximizes the average imputation accuracy.

Imputation accuracy is often assessed through correlation [25,26,27]. Although correlation measures the linear relationship between true and estimated alleles, it does not capture the distance between them. More advanced methods, such as the rate of correct input genotypes (identity percentage), relative Manhattan distance [28], and corrections for posterior genotype probabilities [29], provide more nuanced assessments of accuracy. The relative Manhattan distance is particularly useful, as it accounts for different error types, including those involving correct allele input (e.g., A/T input as A/A or T/T). As shown in this study, the percent identity method is selected for its simplicity and direct comparison of genotypes, although it results in lower substitution accuracy. This method is consistent with the findings of most studies and is better suited for evaluating imputation accuracy.

In practical applications, imputation with a large reference panel often takes several days of computation. KBeagle improves upon this by clustering the population into subgroups based on genetic relationships, allowing subpopulations to share longer haplotype segments than the entire population. Having longer haplotypes means there will be a smaller number of haplotypes. Clustering and multithreaded technology also reduce computational complexity. For Beagle, time complexity is expressed as T = M * N, where T is the computation time, M is the number of individuals, and N is the number of haplotypes estimated from an entire population in Beagle. For KBeagle, it is expressed as T = M * (N_1_ + N_2_ + … + N_i_), where N1…Ni is the number of haplotypes estimated from each subpopulation in KBeagle after the clustered whole population has been divided into subpopulations. That means the length of the haplotypes in the subpopulation will be longer than in the whole population. That also means the number of the haplotypes in the subpopulation will be lower than in the whole population; hence, N > (N_1_ + N_2_ + … + N_i_). According to Ehsan Ullah et al. [30], the GIGI2 imputation method significantly reduces computation time through multithreading. Similarly, compared to Beagle, KBeagle achieves a 2- to 4-fold reduction in computation time by leveraging multithreading as well.

We also compared the results of genomic prediction (GP) and Genome-Wide Association Studies (GWASs) for three datasets: ND, BID, and KID. While the results of the GWASs showed significant differences across datasets, the GP results remained relatively consistent. This can be attributed to factors such as population structure [30], population size, training population composition [31], and marker density [32]. When comparing the datasets from a structural perspective, we found that KBeagle outperforms Beagle in datasets with complex population structures, detecting more SNPs associated with specific traits in Manhattan plots. However, further research is needed to understand why KBeagle does not improve GP accuracy. Previous studies [33] have suggested that performing imputation before quality control can slightly improve GP accuracy, but it may have minimal effects on GP when the dataset has already been quality-controlled. In our study, quality control removed SNPs with low call rates or minor allele frequencies, which may have reduced the dataset’s accuracy. Despite this, KBeagle improved the imputation matching rate, enabling the identification of more SNPs associated with specified traits.

In summary, KBeagle offers faster computation times and better imputation accuracy in distant and complex datasets compared to Beagle. While KID and BID exhibit similar heritability and low prediction accuracy in GP, KID performs better in datasets with rich structures, revealing more trait-associated SNPs. This study underscores the potential of KBeagle as an improved genotype imputation method, which has significant implications for future genomic studies, especially those involving large populations.

## 4. Materials and Methods

### 4.1. Genotype Data and Its Preprocessing

To test KBeagle on a species with relatively complex social structure and natural mating, we used two datasets from previous studies. One includes 80 yaks of 3 breeds, including Maiwa, Yushu, and Huanhu populations with 2,060,483 SNPs (data accessed on 13 January 2021, available at https://doi.org/10.1186/s12864-024-09993-7) [34]. To further analysis, we used the GAPIT package to convert data in hapmap format into data in numeric format serving as the original genotype file (2 M). The other comprised 354 Ashidan yaks with 98,688 markers, of which 75,829 SNPs have known genotypes, while 22,859 SNPs include missing genotypes (data accessed on 13 July 2024, available at https://doi.org/10.1111/age.12897) [35]. We obtained data in the ped file format and used PLINK 1.9 to convert ped files into numeric format files.

Genotypes were coded as 0, 1, and 2, corresponding to major alleles, heterozygous genotype, and minor alleles, respectively. Then, the imputation software, Beagle, was used to impute the converted numerical file, serving as the original genotype file for later experiments. For subsequent comparisons of imputation effects, we maintained the original sample sizes while creating four SNP subsets of varying densities (in which we randomly sampled 10 K, 30 K, 50 K, and 70 K SNP from the whole original genotype dataset) to serve as complete datasets. Subsequently, we set four missingness rates (5%, 10%, 15%, and 20%) to generate missing datasets. Since the data containing NA was not handled by K-Means clustering, we replaced missing values “NA” with “1” in the missing datasets to create corresponding clustering datasets. This entire process from SNP subset generation to missing data introduction was repeated 30 times to ensure robustness, resulting in three perfectly matched dataset types for each iteration: non-missing datasets (with original genotypes), missing datasets (with missing genotypes), and clustering datasets (with imputation placeholder values).

The silhouette coefficient method used to determine the optimal number of clusters, KN. For each k from 2 to 10, we performed K-Means clustering and repeated nstart = 25 times to avoid local optima. Using the silhouette function, we calculated the silhouette coefficients for the current clustering result and stored the mean of all samples’ silhouette coefficients. The k with the highest mean silhouette coefficient was selected as the optimal number of clusters, KN.

### 4.2. Adapter Construction

The input file format is VCF in Beagle 5.4 (Brian L. Browning, Department of Medicine, University of Washington, Seattle, WA, USA.), while both our original files and downstream analysis files need to be in numerical format. So the format of sub-datasets was repeatedly converted from VCF, hapmap, to numeric format among clustering and imputation. To facilitate seamless data conversion throughout our pipeline, we developed a custom adapter that first converts numeric genotypes (0/1/2) into hapmap format by encoding them as “AA”/”AG”/”GG” and marking missing values as “NN”. The hapmap-formatted data was then converted to VCF format using TASSEL v5.0, after which genotype imputation was performed using Beagle v5.4. The imputed VCF results were subsequently converted back to hapmap format using our Perl script vcf2hpm2.pl, and finally the “AA”, ”AG”, ”GG” genotypes were recoded into numeric 0/1/2 format for downstream analysis. The complete workflow is illustrated in Appendix A, and all scripts are publicly available on GitHub (https://github.com/99Xingyu-Guo/KBeagle version 2.0, accessed on 2 June 2025). In the Beagle test, the single thread was used to impute missing genotypes from a whole population. In the KBeagle test, the whole population with missing genotypes was divided into several subpopulations. Then all of the subpopulations simultaneously underwent multithreaded Beagle imputation. In each subpopulation imputation, a single thread was used, as was the case for Beagle.

### 4.3. Beagle

The newest Beagle (version 5.4) software depends on HMM to impute the genotypes of target samples. The Beagle HMM is composed of three processes: state initial probability, state transition probability, and output emission probability. In the model of Li and Stephens [36], the HMM state space is a matrix comprising the alleles of the reference panel. Each row of the matrix represents different haplotypes, and each column represents different markers. The HMM status of a given marker is determined via referring to alleles in the reference panel. The estimated probability of alleles per marker was the summed HMM states of all alleles for that marker [7]. The probability of a specific haplotype was calculated by adding probabilities from the first to the last marker. The haplotype set was obtained via genotyping target samples. The model was established according to haplotypes that were common between the target sample and the reference panel. Haplotype length and number were estimated from the whole population in the HMM, and depended on imputation accuracy and time.

### 4.4. KBeagle

In KBeagle, the number of whole individuals with a missing genotype was initially estimated based on their genetic distance. Based on the genetic distance, the whole population was divided into several clustered groups. The K-Means algorithm was used to perform the clustering. Then the genotypes of individuals in each clustered group were imputed using the Beagle software. Finally, the results of imputation were ordered as described previously. The whole KBeagle process consisted of six steps: data inputting, K-Means clustering, clustering result, imputing, imputing result, and data outputting (Figure 4). Numeric data formats (0, 1, 2, and NA) were accepted as input data. The entire dataset should include individual ID, known genotypes, and genotypes of missing values. Homozygotes of major and minor alleles, as well as heterozygotes, were coded as 0, 2, and 1, respectively (Figure 4A). All markers were used to estimate the genetic distance between each sample. The K-Means algorithm was used to select K cluster centers, calculate the distance between cluster centers and the samples, and group individuals close to the same cluster center into one class. All samples were successfully clustered into K classes (Figure 4B). Samples were grouped together based on clustering results, and K datasets were generated for subsequent imputation (Figure 4C). In converting an adapter, Beagle was used to construct a shared haplotype which refers to highly similar regions within the same cluster between the reference panel and target samples in the clusters in each set, respectively. Then multithreading was applied and the steps were repeated to obtain the K clustering results (Figure 4D). Next, K datasets containing complete genotypes were outputted (Figure 4E). Finally, the results were reordered according to individual taxa, in accordance with the original input data (Figure 4F).

KBeagle, including its code and demo data, is available on Github (https://github.com/99Xingyu-Guo/KBeagle) (accessed on 2 June 2025). The beagle.jar file of Beagle software and testing data are also available via this link.

### 4.5. Imputation Performance Estimation

In this study, we used the imputation matching rate and computation time to evaluate the imputation performances of both Beagle and KBeagle. The imputation matching rate was defined as the number of correctly imputed genotypes divided by the total number of imputed genotypes, calculated as follows:$$R_{\mathrm{match}}=\frac{N_{\mathrm{correct}}}{N_{\mathrm{total}}}$$
where $N_{\mathrm{correct}}\;$ is the total number of accurately predicted genotypes after imputation, and $N_{\mathrm{total}}$ is the total number of all missing genotypes.

Computation time was defined as the latency from the beginning to the end of imputation, including data inputting, K-Means clustering, genotype imputation, and calculation of imputation accuracy. The system.time function began measuring time from the data input step and stopped when the imputation matching rate was calculated.

### 4.6. Genome-Wide Association Study with Imputed Genotype

In the GWAS analysis, we employed the GAPIT [37] package to evaluate detectability with imputed genotype data by assessing the FDR and statistic power for three file types: ND, BID, and KID. Initially, we simulated the phenotype using NDs of varying densities, including 10 K, 30 K, 50 K, and 70 K SNPs. The simulation assumed a heritability (h^2^) with 0.75 and incorporated 20 QTN to generate the phenotypic data. The positions of setting QTNs were recorded for calculation of detected Power and FDR. We directly used the specific names, chromosome numbers, and positions of SNPs in the structured data of three breeds. All genotype data in the homogeneous Ashidan yak population were filtered and mapped with a scaffold-level reference genome (un-chromosome-level reference genome). We integrated all SNPS into a common chromosome named 1. The BLINK model in the GAPIT package was used to perform the GWAS step with three genotype files (ND, BID, and KID), respectively. The Power was defined as the detectability of all setting QTNs. Each QTN provided 5% Power (total 100%, and 20 QTNs). The FDR was defined as the ratio between the number of false-positive signals recorded when GWAS detected for each QTN and the total number of non-QTNs. The type I error was defined as the ratio between the number of false-positive signals when GWAS detected for each QTN setting and the total number of markers. The Power, FDR, and Type I error were recorded after each GWAS program. This process was repeated 60 times and the average was used to define the final results for each dataset.

### 4.7. Genomic Prediction with Imputed Genotype

To validate the practicability of KID, GP mainly including heritability and prediction accuracy were performed using genomic best linear unbiased prediction (GBLUP) [38], 5-fold cross-validation and GAPIT package.

Estimated heritability refers specifically to narrow sense heritability (*h*^2^), representing the proportion of phenotypic variance attributable to additive genetic effects. This key parameter was calculated using GBLUP models and serves as a fundamental metric for evaluating the genetic architecture of traits. It was defined as the rate of additive genetic variance (*V_A_*) in total phenotypic variance (*V_P_*), represented by the following formula [39]:$$h^{2}=\frac{V_{A}}{V_{p}}=\frac{V_{A}}{V_{A}+V_{E}}$$

Simulated phenotype refers to artificially generated phenotype data created using computational models that follow predefined genetic architectures. These simulated datasets are commonly used to evaluate method performance, assess statistical power, or compare algorithms under controlled conditions. The phenotype was simulated using GAPIT with prespecified heritability (*h*^2^ = 0.75), 10 QTNs with effects drawn from a standard normal distribution, and residual variance components calculated based on the heritability ratio. ND, BID, and KID were individually used to estimate heritability in GBLUP. The calculations used to determine the simulated phenotype and estimated heritability were repeated 30 times, and the average value of these replicates was defined as the final estimated heritability for each set.

Prediction accuracy was defined as the Pearson correlation coefficient between the predicted and true phenotypic values. Initially, based on the presupposed heritability of 0.75 and a total of 10 QTNs, all individual phenotypes were simulated. A 5-fold cross-validation was employed to evaluate the performance differences of NA, BID, and KID in GP. After loading all four datasets, including NA, BID, KID, and missing data, the simulated phenotypic data and the samples were randomly divided into five equal subsets. In each validation round, one subset was sequentially selected as the test set (with phenotypic values set to NA) for prediction, while the remaining four subsets were used to train the GBLUP model. Each of the five subsets was used as the test set for prediction. The process was repeated 30 times and the average was used to obtain the final accuracy for each dataset.

## Figures and Tables

**Figure 1 ijms-26-05797-f001:**
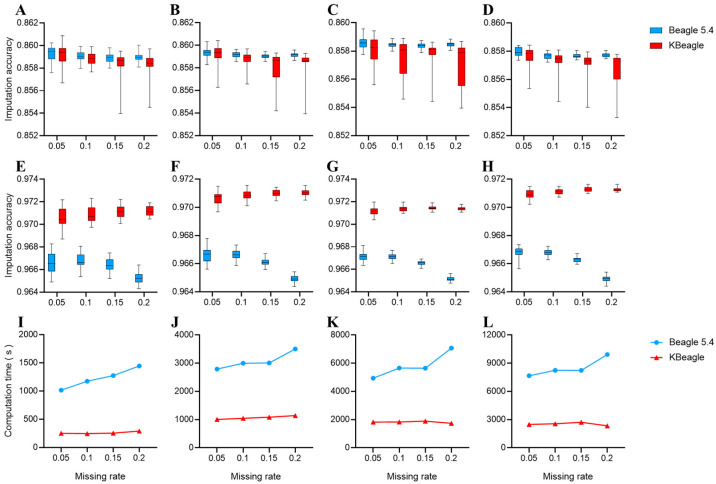
Imputation accuracy and computation time under two methods. (**A**–**H**) Imputation accuracy across different SNP panels: (**A**–**D**) Results for 354 Ashidan yaks (homogeneous data); (**E**–**H**) results for 80 yaks from Maiwa, Yushu, and Huanhu populations (structured data). Each panel represents different SNP densities (10 K, 30 K, 50 K, 70 K from left to right). (**I**–**L**) Computation time comparisons for corresponding datasets. Red line indicates KBeagle; blue line indicates Beagle. All panels display results from multiple replicates as boxplots.

**Figure 2 ijms-26-05797-f002:**
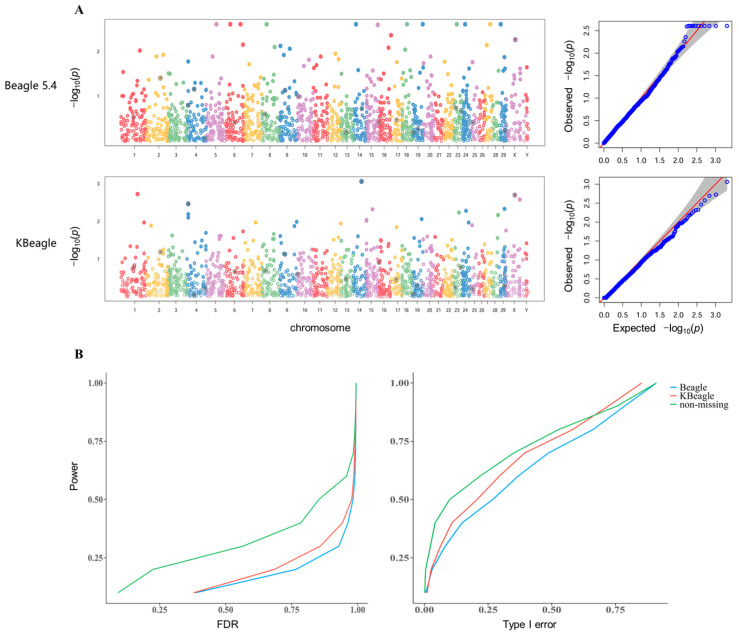
Genome-wide association study of three types of genotype datasets. (**A**) The Manhattan plots and QQ plots indicated the distribution of SNP sites of BID (**above**) and KID (**below**). (**B**) The plot showed the average FDR (**left**) and the average type I error (**right**) of three types of genotype datasets, including ND, as well as BID and KID, under the same power. Green line indicates non-missing, redline indicates KBeagle and blue line indicates Beagle.

**Figure 3 ijms-26-05797-f003:**
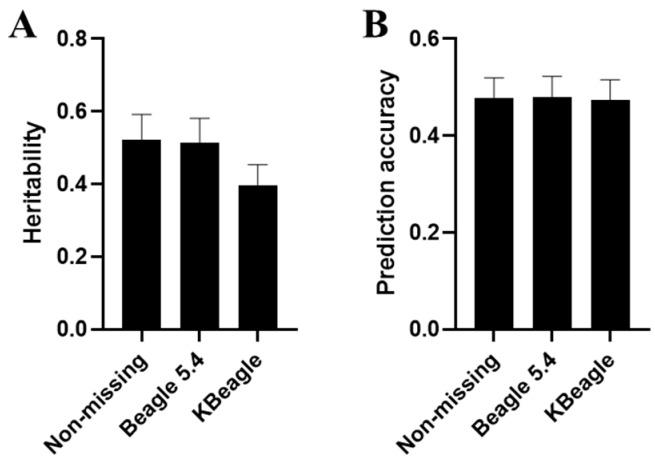
Genomic prediction under three types of genotype datasets. In genomic prediction, the study primarily compared the estimated heritability (**A**) and prediction accuracy (**B**) of ND, as well as BID and KID with the same missing rate (0.1) under 10 K.

**Figure 4 ijms-26-05797-f004:**
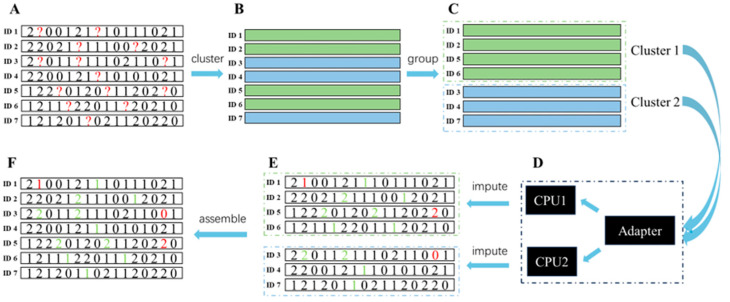
The pipeline of data flow in the K-means Beagle (KBeagle). The whole KBeagle progress comprised 6 steps: (**A**) data inputting; (**B**) K-means clustering; (**C**) clustering result; (**D**) imputing; (**E**) imputing result; (**F**) data outputting. The red “?” marks indicate missing (NA) positions. The red and green numbers represent error imputation and correct imputations, respectively.

## Data Availability

The software packages KBeagle are available from the Github website (https://github.com/99Xingyu-Guo/KBeagle) (accessed on 2 June 2025). The simulated data used in this study are available upon request. The real data used in this study are available at https://www.ncbi.nlm.nih.gov/bioproject/PRJNA899924/ (accessed on 13 January 2021) and available at https://www.animalgenome.org/repository/pub/NWAU2019.0430/ (accessed on 13 July 2024).

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
