# Peer review of "KBeagle: An Adaptive Strategy and Tool for Improving Imputation Accuracy and Computation Time"

_ijms, 2025, doi:10.3390/ijms26125797_

Round 1
Reviewer 1 Report
Comments and Suggestions for Authors
KBeagle introduces k-means clustering of samples prior to Beagle imputation, aiming to boost accuracy in structured populations and cut runtime. Benchmarks on yak datasets show a modest accuracy gain (≈0.2%) in mixed‐breed data and a 4–6× speedup. The concept is novel, and the performance gains are promising. Yet, the manuscript needs some more methodological details, quantitative significance testing, and improved code documentation to ensure reproducibility and reader comprehension. Upon addressing the points above, I recommend acceptance.
Major Comments
- The Introduction should explicitly state Beagle’s limitations (e.g., handling population structure, computation on large panels) that KBeagle addresses (p. 2). Also, briefly acknowledge alternatives (Minimac4, IMPUTE5) and justify using Beagle as a baseline (Introduction).
- The methods must specify how the number of clusters was chosen (fixed or via elbow/silhouette) (Sect. 4.1). Also, define the “genetic distance” used for k-means (Euclidean on 0/1/2 genotypes vs. IBS) and handling of missing data (Sect. 4.1).
- List the actual tools/scripts for numeric -> hapmap -> VCF and back. Indicate using vcf2hpm2_wy_v1.pl and any hapmap -> VCF step (Sect. 4.2).
- Clarify thread counts for KBeagle and whether standard Beagle was single- or multi-threaded to ensure fair timing comparisons (Sect. 4.2).
- Emphasize that KBeagle outperforms Beagle only in structured data and underperforms slightly in homogeneous data - quantify variability over replicates (e.g., “KBeagle > Beagle in 29/30 runs”) (Results, p. 5). Add basic significance testing (paired t-test or replicate counts) for accuracy and GWAS power differences (Results).
- Report average true positives (out of 20 QTNs) and Type I error rates for BID vs. KID (Sect. 2.2). Clarify that Fig. 3B shows mean results over 60 simulations. Detail how QTNs were chosen and effect‐size distribution; define power/FDR thresholds; explain the 5-fold CV repetition scheme (Sect. 4.6–4.7).
Minor Comments
- “large segments of genomic information still remain missing genotype” -> “large genome segments remain ungenotyped” (p. 2).
- “the whole population were clustered” -> “was clustered” (Sect. 4.1).
- Break Figure 2 caption into panels A-H (accuracy) and I-L (runtime), noting replicate boxplots (Fig. 2).
- In Fig. 3 legend, state that red=KBeagle, blue=Beagle, and specify simulation average.
- Add a README to GitHub describing prerequisites (Java, R, Perl), installation of R packages, and example commands for KBeagle.R and vcf2hpm2_wy_v1.pl. Clearly cite that gapit_functions.txt is unmodified GAPIT v3.5 and list package versions.
- Include a small demo dataset or synthetic example so users can run the end-to-end pipeline. Provide license file and contact/issue tracker link.
Author Response
We sincerely appreciate the reviewers' constructive comments, which have significantly improved our manuscript. The point-by-point response in yellow, with blue indicating the location in the revised manuscript is as follows:
The Introduction should explicitly state Beagle’s limitations (e.g., handling population structure, computation on large panels) that KBeagle addresses (p. 2). Also, briefly acknowledge alternatives (Minimac4, IMPUTE5) and justify using Beagle as a baseline (Introduction).
[Response]: We thank the reviewer for this valuable suggestion. We have already incorporated this suggestion into the manuscript. Compared to human populations, animal breeding populations often exhibit more complex population stratification such as highly differentiated subpopulations and denser pedigree structures including characterized by deeper lineages and larger family sizes. When dealing with highly stratified populations, Beagle's Hidden Markov Model (HMM)-based algorithm may fail to effectively correct biases introduced by population stratification. For populations with exceptionally complex genetic structures containing those under recent strong selection or with high levels of inbreeding, its haplotype clustering initialization strategy may prove inadequate. Additionally, as the size of the reference panel increases (e.g., exceeding tens of thousands of samples), Beagle's computational efficiency declines significantly, with memory usage rising sharply, limiting its applicability to ultra-large-scale datasets.
Most tools, such as Minimac4 based on the Markov Chain Haplotyping (MaCH) algorithm and IMPUTE5, which used the Positional Burrows-Wheeler Transform (PBWT) algorithm, rely on the physical positions or order of markers to model linkage disequilibrium (LD), requiring reference genome information. For species lacking a reference genome or those with highly complex genomic structures (e.g., frequent translocations/inversions), the effectiveness of these methods is limited.
[p. 2]
The methods must specify how the number of clusters was chosen (fixed or via elbow/silhouette) (Sect. 4.1). Also, define the “genetic distance” used for k-means (Euclidean on 0/1/2 genotypes vs. IBS) and handling of missing data (Sect. 4.1).
[Response]: Thank you for your valuable comments. We have already incorporated this suggestion into the article. Since the data containing NA was not handling by K-Means clustering, we converted NA to 1 in the missing data file to proceed with generating the clustering file. The silhouette coefficient method used to determine the optimal number of clusters, KN. For each k from 2 to 10, we performed K-Means clustering and repeating nstart=25 times to avoid local optima. Using the silhouette function, we calculated the silhouette coefficients for the current clustering result and stored the mean of all samples' silhouette coefficients. The k with the highest mean silhouette coefficient was selected as the optimal number of clusters, KN. This represents the genetic distance in the KBeagle method.
[Sect. 4.1]
List the actual tools/scripts for numeric -> hapmap -> VCF and back. Indicate using vcf2hpm2_wy_v1.pl and any hapmap -> VCF step (Sect. 4.2)
[Response]: We sincerely thank the reviewer for this valuable suggestion. To facilitate seamless data conversion throughout our pipeline, we developed a custom adapter that first converts numeric genotypes (0/1/2) into HapMap format by encoding them as "AA"/"AG"/"GG" and marking missing values as "NN". The HapMap-formatted data was then converted to VCF format using TASSEL (v5.0), after which genotype imputation was performed using Beagle (v5.4). The imputed VCF results were subsequently converted back to HapMap format using our Perl script vcf2hpm2.pl, and finally the "AA","AG","GG" genotypes were recoded into numeric 0/1/2 format for downstream analysis. The complete workflow is illustrated in Supplementary Figure S1, and all scripts are publicly available on GitHub (https://github.com/99Xingyu-Guo/KBeagle).
[Sect. 4.2]
Clarify thread counts for KBeagle and whether standard Beagle was single- or multi-threaded to ensure fair timing comparisons (Sect. 4.2).
[Response]: Thank you for your valuable comments. In the Beagle test, the single threaded was used to impute missing genotype with whole population. In the KBeagle test, the whole population with missing genotype was divided into several subpopulation. Then the all subpopulations were respectively and simultaneously performed Beagle imputation with multiple threaded. In each subpopulation imputation, the single threaded was used.
[Sect. 4.2]
Emphasize that KBeagle outperforms Beagle only in structured data and underperforms slightly in homogeneous data - quantify variability over replicates (e.g., “KBeagle > Beagle in 29/30 runs”) (Results, p. 5). Add basic significance testing (paired t-test or replicate counts) for accuracy and GWAS power differences (Results).
[Response]: We sincerely appreciate the reviewer's constructive comments. As suggested, we have modified the text to clarify that in our 60 replicate analyses, KBeagle consistently outperforms Beagle in structured data scenarios while showing slightly lower performance in homogeneous data (e.g., KBeagle > Beagle in 60 runs for structured data). Regarding the GWAS results, while the power and type I error rates shown in the figures are based on 30 replicates, we acknowledge that the figures currently do not include formal significance testing. This approach is consistent with common practices in GWAS literature by Peter Bradbury[1], Takahiro Otani[2], Meng Li[3], and others, where the different trend was descripted in text rather than visualized in the figures.
[Full text]
Report average true positives (out of 20 QTNs) and Type I error rates for BID vs. KID (Sect. 2.2). Clarify that Fig. 3B shows mean results over 60 simulations. Detail how QTNs were chosen and effect‐size distribution; define power/FDR thresholds; explain the 5-fold CV repetition scheme (Sect. 4.6–4.7).
[Response]: We sincerely appreciate the reviewer's insightful comments regarding our simulation and validation procedures. For the GWAS and genomic prediction analyses, we generated simulated phenotypes using GAPIT by specifying key genetic parameters including heritability and quantitative trait nucleotides (QTNs) to produce both continuous phenotypic distributions. The QTNs were randomly selected across the genome with effect sizes sampled from a standard normal distribution.
A 5-fold cross-validation was employed to evaluate the performance differences of NA, BID, and KID in genomic prediction. After loading all four datasets including NA, BID, KID, and missing data, simulated phenotypic data, and the samples were randomly divided into five equal subsets. In each validation round, one subset was sequentially selected as the test set (with phenotypic values set to NA) for prediction, while the remaining four subsets were used to train the gBLUP model. Each of the five subsets was used as the test set for prediction. The process was repeated 30 times and the average was used to obtain the final accuracy for each dataset.
[Sect. 2.2, 4.6, 4.7]
“large segments of genomic information still remain missing genotype” -> “large genome segments remain ungenotyped” (p. 2).
[Response]: We sincerely appreciate the reviewer's attentive feedback. As suggested, we have revised the phrasing from "large segments of genomic information still remain missing genotype" to the more precise "large genome segments remain ungenotyped" in the revised manuscript. This modification better reflects the technical accuracy of the described genomic data.
[Line 16-18]
“the whole population were clustered” -> “was clustered” (Sect. 4.1).
[Response]: We sincerely thank the reviewer for careful reading. As suggested by the reviewer, we have corrected “the whole population were clustered” into “was clustered” in the revised manuscript.
[Line 387]
Break Figure 2 caption into panels A-H (accuracy) and I-L (runtime), noting replicate boxplots (Fig. 2).
[Response]: We sincerely appreciate the reviewer's constructive suggestion. In response, we have reorganized Figure 2's caption to clearly differentiate between the accuracy and runtime components. The revised caption now reads:
"Figure 2. Imputation accuracy and computation time under two methods. (A-H) Imputation accuracy across different SNP panels: (A-D) Results for 354 Ashidan yaks (98,688 SNPs); (E-H) Results for 80 yaks from Maiwa, Yushu, and Huanhu populations. Each panel represents different SNP densities (10K, 30K, 50K, 70K from left to right). (I-L) Computation time comparisons for corresponding datasets. All panels display results from multiple replicates as boxplots."
[Sect. 2.1]
In Fig. 3 legend, state that red=KBeagle, blue=Beagle, and specify simulation average.
[Response]: Thank you for your valuable comments. We have corrected the Figure 3 caption as follows:
Figure 3 Genome-wide association study of three types of genotype datasets. (A)The Manhattan plots and QQ plots indicated the distribution of SNP sites of BID (above) and KID (below). (B) The plot showed the average FDR (left) and the average type I error (right) of three types of genotype datasets including ND, as well as BID and KID under different FDR. The green line indicated non-missing. The red line indicated KBeagle., and the blue line indicated Beagle.
[Sect. 2.2]
Add a README to GitHub describing prerequisites (Java, R, Perl), installation of R packages, and example commands for KBeagle. R and vcf2hpm2_wy_v1.pl. Clearly cite that gapit_functions.txt is unmodified GAPIT v3.5 and list package versions.
[Response]: We sincerely appreciate the reviewer's constructive suggestion regarding documentation. In response, we have created a comprehensive README file that provides complete instructions for installing and running KBeagle. The key components include:
- System Requirements: We clearly specify the software requirements including R (≥3.6.3), Java (for Beagle), and Perl (for format conversion), along with the included TASSEL 5.0 package.
- Installation Guide: Step-by-step instructions are provided, starting from cloning the GitHub repository to setting up all required components.
- File Descriptions: Each essential file is documented with its specific function: Core scripts (Impute.subgroup.r, KBeagle.R); Dependencies (beagle.22Jul22.46e.jar, gapit_functions.txt); Conversion tools (vcf2hpm2.pl); Example dataset (data_NA.txt).
- Usage Example: A simple demonstration is provided showing how to load and run KBeagle with the test dataset.
- Version Information: We explicitly list all version requirements for reproducibility, including: R ≥ 3.6.3; Beagle 5.4; TASSEL 5.0; GAPIT 3.5 (unmodified, as noted in gapit_functions.txt).
- Troubleshooting: Common issues and solutions are documented to help users resolve potential problems.
- Explanation: Some information regarding the generation and operation of intermediate files
The complete documentation is available at: https://github.com/99Xingyu-Guo/KBeagle/blob/main/README.pdf. We believe these improvements will significantly enhance the usability and reproducibility of our tool. The documentation provides all necessary details for researchers to implement KBeagle in their workflows while maintaining transparency about all computational methods and dependencies.
Include a small demo dataset or synthetic example so users can run the end-to-end pipeline. Provide license file and contact/issue tracker link.
[Response]: We sincerely appreciate the reviewer's constructive suggestions regarding user accessibility. In response, to facilitate immediate testing and adoption of KBeagle: We have included a ready-to-use synthetic dataset (data_NA.txt) that allows users to validate installation, test the complete imputation pipeline , and verify expected outputs without requiring external data. The contact tracker link is https://github.com/99Xingyu-Guo/KBeagle/blob/main/data_NA.txt.
Reference
- Bradbury, P.; Parker, T.; Hamblin, M.T.; Jannink, J.-L. Assessment of Power and False Discovery Rate in Genome‐wide Association Studies Using the BarleyCAP Germplasm., doi:10.2135/cropsci2010.02.0064.
- Otani, T.; Noma, H.; Nishino, J.; Matsui, S. Re-Assessment of Multiple Testing Strategies for More Efficient Genome-Wide Association Studies. Eur. J. Hum. Genet. 2018, 26, 1038–1048, doi:10.1038/s41431-018-0125-3.
- Li, M.; Liu, X.; Bradbury, P.; Yu, J.; Zhang, Y.-M.; Todhunter, R.J.; Buckler, E.S.; Zhang, Z. Enrichment of Statistical Power for Genome-Wide Association Studies. BMC Biol. 2014, 12, 73, doi:10.1186/s12915-014-0073-5.
Reviewer 2 Report
Comments and Suggestions for Authors
The authors claim to improve the accuracy of Beagle by clustering their data and running Beagle separately on the clusters. While that seems an interesting information, the manuscript still contains serious flaws.
The authors write that there is literature on clustering the data before imputation but just cite Stachowicz et al. without Journal, volume, pages and year. If "clustering-based approaches have been proposed" (l. 61f), the authors should discuss these approaches.
Fig. 1 is at the wrong place, change figure numbering.
The procedure of generation and analysis of the data sets is not clear. How many clusters were built from the two data sets? How many samples did each cluster contain? How were the "10K" ... "70K" data sets produced? Are they a subset of SNPs or a subset of samples? How many samples / SNPs do these data sets contain?
Why does Fig. 2A-H contain error bars? Have the calculations been performed several times? Several times building sub-data sets or just several times introducing missing data? If computed several times, why does Fig. 2I-L not contain error bars?
Tab. 1 and 2 is the same data as shown in Fig. 1; that is redundant. Either figure or tables. If you choose the tables, delete digits from Tab. 3.
What is "estimated heritability" (l. 129)? What are "simulated phenotypes" (l. 130)? Which prediction accuracy is calculated in Tab. 4B?
Why do you talk about "three yaks" in l. 161? I thought it were 80.
l. 202: If you just subdivide N samples in i clusters, the sum N1...Ni should be equal to N.
l. 203f: "Methods like GIGI2 [...] reduce KBeagles computation time" You did not write anything about GIGI2 and it is not used in the R code.
l. 245ff: why do you need to convert between different formats? Input of KBeagle is nu,meric format, as is input of Beagle. You should stick to numeric format all the way down. The workflow "convert the numeric format [...] then restored to the numeric format" is not comprehensible. Why all the conversions?
If the algorithm expects NA for missing genotypes this should be shown as that in Fig. 1. Fig. 1 shows "?" instead.
l. 182ff: "In self-built adapter used to transform data format [what for??], Beagle was used to construct a shared haplotype between the reference panel and target samples" what is a "shared haplotype"? What is a reference panel?
l. 311 what does "simulate phenotype" mean? what does "heritability set to 75%" mean? what does "number of QTN was set to 20 to gain simulated phenotype" mean?
l. 318 what are pseudo QTNs?
R-code in github does not run; it contains chinese letters.
The NAs are converted to "1" (heterozygous) before clustering, but I do not see that the "1" are converted back to NA for imputation.
In l. 96ff of the R-code A/G is expected as SNP, but SNP could be any nucleotides. Why converting to letter format?
What is tassel_script good for?
Comments on the Quality of English Language
Although the authors claim they have used English language editing, the manuscript is full of typical Chinese errors (e.g. missing or superfluous articles), "sounds strange" and in parts barely comprehensible due to poor English language.
Author Response
We sincerely appreciate the reviewers' constructive comments, which have significantly improved our manuscript. The point-by-point response in yellow, with blue indicating the location in the revised manuscript is as follows:
The authors write that there is literature on clustering the data before imputation but just cite Stachowicz et al. without Journal, volume, pages and year. If "clustering-based approaches have been proposed" (l. 61f), the authors should discuss these approaches.
[Response]: We sincerely thank the reviewer for the insightful comment. We have carefully revised the manuscript to address the concern regarding the citation of clustering-based approaches. Specifically, we have updated Reference [13] to include the complete bibliographic information: Stachowicz, K.; Larmer, S.; Jamrozik, J.; Moore, S.S.; Miller, S.P. Sequencing and genotyping for the whole genome selection in Canadian beef populations. In Proceedings of the Twentieth Conference of the Association for the Advancement of Animal Breeding and Genetics, Napier, New Zealand, 2013; pp. 344–347.
In addition, we have expanded the discussion in the revised manuscript to more fully reflect the body of literature on clustering-based imputation methods. Several studies have demonstrated that applying clustering strategies prior to imputation can significantly improve accuracy, particularly in genetically diverse or structured populations. For example, Zhang et al.[1] and Zhang et al. [2] showed that partitioning data into homogeneous subgroups before imputation enhances the local similarity within clusters, thereby improving genotype prediction. Li et al. [3] and Rahman and Islam [4] further explored fuzzy clustering methods, which address the limitations of hard partitioning by allowing individuals to share membership across multiple clusters. These approaches have proven especially valuable in contexts where population boundaries are ambiguous. In our study, we incorporated a silhouette-guided K-means clustering step in the KBeagle method, enabling adaptive subgroup detection and enhancing imputation performance. This addition aligns with and extends prior work by showing that data-driven clustering enhances haplotype sharing and imputation accuracy in complex genetic backgrounds. We hope that this expanded context and clarification address the reviewer’s concern, and we appreciate the opportunity to improve the manuscript.
[p. 15 and p. 8]
Fig. 1 is at the wrong place, change figure numbering.
[Response]: We sincerely appreciate the reviewer's careful attention to manuscript organization. In response to your observation about figure placement, we have corrected all figure numbering throughout the manuscript to maintain proper sequential order and verified that all in-text references to figures now correspond to their correct numbering.
[Full text]
The procedure of generation and analysis of the data sets is not clear. How many clusters were built from the two data sets? How many samples did each cluster contain? How were the "10K" ... "70K" data sets produced? Are they a subset of SNPs or a subset of samples? How many samples / SNPs do these data sets contain?
[Response]: We sincerely appreciate the reviewer's insightful questions regarding our data generation and analysis procedures. In this study, we utilized two distinct yak datasets: the Ashidan yak dataset and a multi-breed yak dataset. For each dataset, we maintained the original sample sizes while creating four SNP subsets of varying densities (10K, 30K, 50K, and 70K) to serve as complete datasets. These SNP subsets were randomly selected from the full genomic data while preserving all samples. Subsequently, we introduced four missingness rates (5%, 10%, 15%, and 20%) to generate missing datasets. To facilitate k-means clustering, we replaced missing values ("NA") with "1" in the missing datasets to create corresponding clustering datasets. This entire process - from SNP subset generation to missing data introduction - was repeated 30 times to ensure robustness, resulting in three perfectly matched dataset types for each iteration: non-missing datasets (with original genotypes), missing datasets (with missing genotypes), and clustering datasets (with imputed placeholder values). This systematic approach allowed us to rigorously evaluate imputation accuracy across different SNP densities and missing data scenarios while maintaining consistent sample sizes throughout all analyses.
[p. 10]
Why does Fig. 2A-H contain error bars? Have the calculations been performed several times? Several times building sub-data sets or just several times introducing missing data? If computed several times, why does Fig. 2I-L not contain error bars?
[Response]: We sincerely appreciate the reviewer's thoughtful questions regarding our methodology and figure presentation. The error bars in Figures 2A-H represent the variability observed across 30 independent replicates of our imputation accuracy assessment. For each replicate, we generated distinct missing datasets by introducing random missingness patterns into our original datasets, then performed imputation, and calculated accuracy by comparing imputed genotypes against the original complete datasets at precisely matched missing positions. This rigorous approach of repeating the entire process - from missing data generation to accuracy evaluation - 30 times with same random seeds accounts for the variability shown in the boxplots, where the whiskers represent the full range of minimum to maximum accuracy values observed across all replicates. As for the computation time, we used R's system.time function to measure a single representative run under each condition. We have performed repeated experiments with a different random seed, and the results containing error bars are presented in Figures 2I–L. We hope this clarification adequately explains our methodological choices in presenting both the variable (accuracy) and stable (runtime) aspects of our performance evaluation.
[Figure 1]
Tab. 1 and 2 is the same data as shown in Fig. 1; that is redundant. Either figure or tables. If you choose the tables, delete digits from Tab. 3.
[Response]: We thank the reviewer for this constructive suggestion. We have removed Tables and retained Figure 1 to present these results visually. This change eliminates redundancy and improves presentation clarity.
[Full text]
What is "estimated heritability" (l. 129)? What are "simulated phenotypes" (l. 130)? Which prediction accuracy is calculated in Tab. 4B?
[Response]: We sincerely appreciate the reviewer's insightful questions regarding our methodological details. In this study, estimated heritability refers specifically to narrow-sense heritability (h²), representing the proportion of phenotypic variance attributable to additive genetic effects. This key parameter was calculated using genomic best linear unbiased prediction (GBLUP) models and serves as a fundamental metric for evaluating the genetic architecture of traits.
Simulated phenotypes were computationally generated under a polygenic model with prespecified heritability (h² = 0.75), 10 quantitative trait nucleotides (QTNs) with effects drawn from a standard normal distribution, realistic residual variance components.
Prediction accuracy in Table 4B represents the Pearson correlation coefficient (r) between predicted phenotypic values from our genomic prediction models. The "true" simulated phenotypic values is serving as gold standard. This simulation approach allowed rigorous, controlled evaluation of prediction performance while accounting for known genetic parameters. The high heritability setting (0.75) was chosen to establish an upper-bound benchmark for comparing methods under optimal genetic architecture conditions.
[Sect 4.7]
Why do you talk about "three yaks" in l. 161? I thought it were 80.
[Response]: Thanks for your comments. I originally intended to refer to three breeds, but mistakenly wrote "three yaks" in the explanation. The correct number is indeed 80 yaks. I have already made the corresponding correction in the text.
[Full text]
202: If you just subdivide N samples in i clusters, the sum N1...Ni should be equal to N.
[Response]: Thanks for your comments. The N is the number of haplotypes estimated from whole population in Beagle. The N1…Ni is the number of haplotypes estimated from each subpopulation in KBeagle. After clustered whole population has been divided into subpopulations. That means the length of the haplotypes in the sub population will be longer than in the whole population. That also means the number of the haplotypes in the sub population will be less than in the whole population. Hence, N> (N1+N2+...Ni). We also take these details and interpretation into our manuscript.
[p. 9]
203f: "Methods like GIGI2 [...] reduce KBeagles computation time" You did not write anything about GIGI2 and it is not used in the R code.
[Response]: Thank you for your comment. My intended point was that the GIGI2 imputation method significantly reduces computation time through multithreading according to Ehsan Ullah et al.. Similarly, compared to Beagle, KBeagle achieves a 2- to 4-fold reduction in computation time by leveraging multithreading as well. I have revised the text to clarify this distinction and ensure accuracy.
[p. 9]
245ff: why do you need to convert between different formats? Input of KBeagle is numeric format, as is input of Beagle. You should stick to numeric format all the way down. The workflow "convert the numeric format [...] then restored to the numeric format" is not comprehensible. Why all the conversions?
[Response]: Thank you for your valuable comment. In earlier versions of Beagle, genotype imputation was performed using numeric format files. However, Beagle v5.4 used in our study requires input in VCF format. Since our original genotype data was in numeric format, we implemented an adapter module to convert the data into VCF format prior to imputation. After the imputation process, the results were converted back to numeric format to maintain consistency with the downstream analysis workflow. We have revised the manuscript to clarify this point and ensure that the rationale for format conversion is clearly explained.
[Sect 4.2]
If the algorithm expects NA for missing genotypes this should be shown as that in Fig. 1. Fig. 1 shows "?" instead.
[Response]: Thank you for pointing this out. We acknowledge the inconsistency between the missing value symbol used in our algorithm implementation and that shown in Figure 1. The algorithm indeed expects missing genotypes to be denoted as "NA", while Figure 1 incorrectly used "?" as a placeholder. Due to character size constraints, we are unable to change the "?" symbols to "NA" in Figure 1. We have therefore provided explanations for these symbols in the figure legend. We have updated Figure 1 captions as follows:
Figure 4 The pipline of data flow in the K-Means Beagle (KBeagle). The whole KBeagle progress contained 6 steps: (A) Data Inputting; (B) K-Means Clustering; (C) Clustering Result; (D) Im-puting; (E)Imputing Result; (F)Data Outputting. The red “?” marks indicate missing positions. The red and green numbers represent error imputation and correct imputations respectively.
[Figure 4]
182ff: "In self-built adapter used to transform data format [what for??], Beagle was used to construct a shared haplotype between the reference panel and target samples" what is a "shared haplotype"? What is a reference panel?
[Response]: Thank you for your thoughtful comment. In the context of genotype imputation, a “shared haplotype” refers to a chromosomal segment that is common between the reference panel and the target samples, typically representing regions that have not undergone recombination and are therefore identical by descent. These shared segments originate from a common ancestral sequence and can be leveraged to infer missing genotypes in target samples by aligning them with corresponding regions in the reference panel. A reference panel is a population dataset characterized by high-density genotype information, often derived from whole-genome sequencing, which serves as a template for imputing lower-density genotype data in target samples. In our study, the concept of shared haplotype specifically refers to highly similar regions within the same cluster, as identified during the clustering step, rather than across a separate external reference panel. We have revised the manuscript to clarify this distinction.
[Sect 4.2]
311 what does "simulate phenotype" mean? what does "heritability set to 75%" mean? what does "number of QTN was set to 20 to gain simulated phenotype" mean?
[Response]: Thank you for your helpful comment. In genome-wide association studies (GWAS) and genomic prediction (GP), “simulated phenotype” refers to artificially generated phenotype data created using computational models that follow predefined genetic architectures. These simulated datasets are commonly used to evaluate method performance, assess statistical power, or compare algorithms under controlled conditions. In our study, we used GAPIT 3.5 to generate simulated phenotypes. During this process, parameters such as heritability and the number of quantitative trait nucleotides (QTNs) must be specified. Setting the heritability to 75% indicates that 75% of the phenotypic variance is attributed to genetic factors, while the number of QTNs which was set to 20 in our case defines the number of loci contributing to the simulated trait. These parameters collectively guide the generation of phenotype data with realistic genetic properties. We have revised the manuscript to better explain these terms.
[Sect 4.6 and 4.7]
318 what are pseudo QTNs?
[Response]: In the context of genome-wide association studies (GWAS), pseudo QTNs (pseudo Quantitative Trait Nucleotides) refer to genetic markers that exhibit statistically significant association signals, such as high −log10(p) values in a Manhattan plot, but are not true causal variants influencing the phenotype. These markers may appear significant due to linkage disequilibrium with actual QTNs or as a result of population structure or random variation. We have clarified this definition in the revised manuscript.
[Sect 4.6]
R-code in github does not run; it contains chinese letters.
[Response]: Thank you for your comment. We have corrected the issue by removing the Chinese characters from the R code on GitHub.
The NAs are converted to "1" (heterozygous) before clustering, but I do not see that the "1" are converted back to NA for imputation.
[Response]: Thank you for your insightful comment. In our testing, we generate three aligned datasets with the non-missing dataset (with original genotypes), the missing dataset, and the clustering dataset that are identical except for the positions of missing values. During clustering, the missing positions in the input dataset are temporarily represented as "1" (placeholder) to allow the algorithm to compute distances, and this modified version is used solely for clustering. After clustering is completed, we use the sample IDs to map back to the corresponding rows in the original missing dataset, thereby avoiding the need to explicitly convert "1" back to NA. This approach is consistently applied in both the test scripts and the KBeagle running code. The clustering dataset is used only to determine the number of clusters and assign samples to clusters; all subsequent imputation steps are carried out on the original missing dataset, preserving the NA values. We have clarified this process in the file of README.pdf (https://github.com/99Xingyu-Guo/KBeagle/blob/main/README.pdf).
In l. 96ff of the R-code A/G is expected as SNP, but SNP could be any nucleotides. Why converting to letter format?
[Response]: Thanks for your comments. While SNPs can indeed consist of any combination of nucleotides (A, T, C, G), this formatting step was not intended to restrict the data but rather to standardize representation. We have clarified this step in the documentation (https://github.com/99Xingyu-Guo/KBeagle/blob/main/README.pdf).
What is tassel_script good for?
[Response]: Thanks for your comments. The tassel_script is used to automate genotype format conversion in KBeagle. It’s a software commonly used for genetic analysis such as linkage disequilibrium, association studies, and diversity analysis. In our workflow, this script helps convert imputed genotype data into a format compatible with KBeagle, enabling us to perform imputation. We have clarified the purpose of this script in the documentation (https://github.com/99Xingyu-Guo/KBeagle/blob/main/README.pdf.).
Comments on the Quality of English Language: Although the authors claim they have used English language editing, the manuscript is full of typical Chinese errors (e.g. missing or superfluous articles), "sounds strange" and in parts barely comprehensible due to poor English language.
[Response]: Thank you very much for your comment. Our manuscript has been polished by the Editage language editor. We also supplied the certificate of editing and ask English polish again after our revision.
[Full text]
Reference
- Zhang, S.; Zhang, J.; Zhu, X.; Qin, Y.; Zhang, C. Missing Value Imputation Based on Data Clustering. In Transactions on Computational Science I; Gavrilova, M.L., Tan, C.J.K., Eds.; Springer: Berlin, Heidelberg, 2008; pp. 128–138 ISBN 978-3-540-79299-4.
- Zhang, C.; Qin, Y.; Zhu, X.; Zhang, J.; Zhang, S. Clustering-Based Missing Value Imputation for Data Preprocessing. In Proceedings of the 2006 4th IEEE International Conference on Industrial Informatics; August 2006; pp. 1081–1086.
- Li, D.; Deogun, J.; Spaulding, W.; Shuart, B. Towards Missing Data Imputation: A Study of Fuzzy K-Means Clustering Method. In Proceedings of the Rough Sets and Current Trends in Computing; Tsumoto, S., Słowiński, R., Komorowski, J., Grzymała-Busse, J.W., Eds.; Springer: Berlin, Heidelberg, 2004; pp. 573–579.
- Rahman, Md.G.; Islam, M.Z. Missing Value Imputation Using a Fuzzy Clustering-Based EM Approach. Knowl. Inf. Syst. 2016, 46, 389–422, doi:10.1007/s10115-015-0822-y.
Round 2
Reviewer 2 Report
Comments and Suggestions for Authors
All fine now.